# Comprehensive Analysis of Cuproptosis-Related Genes in Prognosis and Immune Infiltration of Hepatocellular Carcinoma Based on Bulk and Single-Cell RNA Sequencing Data

**DOI:** 10.3390/cancers14225713

**Published:** 2022-11-21

**Authors:** Chenglei Yang, Yanlin Guo, Zongze Wu, Juntao Huang, Bangde Xiang

**Affiliations:** 1Department of Hepatobiliary Surgery, Guangxi Medical University Cancer Hospital, Nanning 530021, China; 2Guangxi Liver Cancer Diagnosis and Treatment Engineering and Technology Research Center, Nanning 530021, China; 3Key Laboratory of Early Prevention and Treatment for Regional High Frequency Tumor, Ministry of Education, Nanning 530021, China; 4School of Basic Medicine, Guangxi Medical University, Nanning 530021, China; 5The First Clinical School of Guangxi Medical University, Nanning 530021, China

**Keywords:** hepatocellular carcinoma, cuproptosis, cuproptosis-related gene signature, immune microenvironment

## Abstract

**Simple Summary:**

Hepatocellular carcinoma (HCC) is a common malignant tumor with high mortality. Cuproptosis is a newly discovered mechanism of cell death. Numerous cuproptosis-related-genes are reported; however, the studies on the role of cuproptosis-related genes in the prognosis of HCC are limited. This study aimed to establish a prognosis model of HCC related to cuproptosis-related genes and explore the correlations between cuproptosis-related genes and the immune microenvironment of HCC. These findings might aid in studying the cuproptosis and tumor immune microenvironment of HCC patients for improving their treatment outcomes and prognosis.

**Abstract:**

Background: Studies on prognostic potential and tumor immune microenvironment (TIME) characteristics of cuproptosis-related genes (CRGs) in hepatocellular carcinoma (HCC) are limited. Methods: A multigene signature model was constructed using the least absolute shrinkage and selection operator (LASSO) Cox regression analysis. The cuproptosis-related multivariate cox regression analysis and bulk RNA-seq-based immune infiltration analysis were performed. The results were verified using two cohorts. The enrichment of CRGs in T cells based on single-cell RNA sequencing (scRNA-seq) was performed. Real-time polymerase chain reaction (RT-PCR) and multiplex immunofluorescence staining were performed to verify the reliability of the conclusions. Results: A four-gene risk scoring model was constructed. Kaplan–Meier curve analysis showed that the high-risk group had a worse prognosis (*p* < 0.001). The time-dependent receiver operating characteristic (ROC) curve showed that the OS risk score prediction performance was good. These results were further confirmed in the validation queue. Meanwhile, the Tregs and macrophages were enriched in the cuproptosis-related TIME of HCC. Conclusions: The CRGs-based signature model could predict the prognosis of HCC. Treg and macrophages were significantly enriched in cuproptosis-related HCC, which was associated with the depletion of proliferating T cells.

## 1. Introduction

Hepatocellular carcinoma (HCC) is a very common malignant tumor with high mortality worldwide. According to the International Agency for Research on Cancer, the Global Cancer Statistics 2020 showed that the number of new cases and deaths caused by HCC worldwide were approximately 900,000 and 800,000, respectively [1]. Patients with HCC are usually diagnosed at an advanced stage [2]. Therefore, given the high morbidity and mortality rates of HCC, effective and accurate prognostic models are needed to be developed for the early detection of HCC.

Copper is an essential element for the biological activities of organisms, as well as a cofactor for various metabolic enzymes. An increase in the concentration of intracellular copper was closely associated with the progression of cancer and was required for the characteristic phenomena, such as angiogenesis and metastasis in the progression of cancer; however, the mechanisms are unknown [3]. Recently, a new cell-death mode, known as cuproptosis, was discovered, which was different from the known cell-death modes [4] and might be an important mechanism for the effects of copper concentration on cancer progression. Cuproptosis is a novel form of copper-dependent cell death that depends on mitochondrial respiration and is associated with the tricarboxylic acid (TCA) cycle [4]. Therefore, cuproptosis might be closely related to the intracellular concentration of copper. The change in the intracellular concentration of copper could affect the occurrence of cuproptosis and significantly affect the occurrence and development of HCC. A previous study identified some cuproptosis-related genes (CRGs); these genes could be used for developing a model by analyzing their expression levels to predict the prognosis of HCC patients [4].

Tumor is not only a combination of malignantly proliferating cells but also a complex tissue consisting of well-organized different cell types [5]. The immune components of the tumor tissues, known as the tumor immune microenvironment (TIME), include tumor immune cells and factors [6,7,8]. TIME is associated with the occurrence, development, prognosis, and recurrence of tumors [9]. T-cell-mediated cellular immunity, playing a dominant role in tumor immunity, can be divided into CD4+ T cells, CD8+ T cells, Treg cells, etc. The CD4+ T cells can differentiate into various subtypes, which play important roles in the coordination of various immune responses in TIME. The CD8+ T cells can target and destroy tumor cells by detecting abnormal tumor antigens expressed on the surface of cancer cells and are usually associated with the efficacy and survival of cancer patients. The Treg cells are ubiquitous in TIME and promote the development and progression of the tumor by suppressing the anti-tumor immune responses, such as the secretion of interleukin-2 (IL-2) [10].

In this study, prognostic models were established, and the TIME characteristics of 361 HCC tissue samples were analyzed using the RNA-seq data obtained from The Cancer Genome Atlas (TCGA) database. The results were further validated using the RNA-seq data of 231 HCC samples from the International Cancer Genome Consortium (ICGC) (LIRI-JP) cohort and 116 HCC samples obtained from our in-house cohort. In addition, the TIME of HCC cuproptosis was further elucidated at the single-cell level. The schematic process of this study was showed in the Figure 1. This study revealed a potential correlation between cuproptosis, prognosis, and TIME in HCC patients. These findings might aid in studying cuproptosis and TIME of HCC patients for improving their treatment outcomes and prognosis.

## 2. Materials and Methods

### 2.1. Dataset Acquisition

On 20 November 2021, the sequencing data and clinical information of 424 patients were obtained from the TCGA database. The data of the patients, who were histopathologically diagnosed with fibrolamellar carcinoma or mixed HCC, were excluded, resulting in a total of 411 patients (361 HCC tumor tissues and 50 non-tumor tissues) (https://portal.gdc.cancer.gov/repository, accessed on 14 January 2022). Moreover, the results were validated using the data of 231 HCC patients obtained from the ICGC database (https://dcc.icgc.org/, accessed on 1 June 2022) and 116 HCC patients from our in-house cohort. The clinical data of all the patients in the three cohorts are listed in Table 1.

Moreover, 19 CRGs were obtained from the published studies [4] (Table 2).

### 2.2. Identification of Prognostic CRGs and Selection of Differentially Expressed Genes Related to Cuproptosis (CRGs-DEGs)

The DEGs were obtained by differential analysis of the HCC tumor and non-tumor samples using the R package “limma” in the TCGA cohort (*p* < 0.05). At the same time, univariate Cox regression analysis was applied to 19 CRGs, which resulted in obtaining 9 prognosis-related CRGs. A total of 16 CRGs-DEGs were obtained by intersecting the DEGs with 19 CRGs obtained from the published studies. The 16 CRGs-DEGs intersected with 9 prognosis-related CRGs, which identified 9 prognosis-related genes as CRGs-DEGs. Therefore 9 CRGs-DEGs related to prognosis were finally obtained and used for subsequent analysis (Table 2).

### 2.3. Development and Validation of the Prediction Model

In order to avoid overfitting, the R package “glmnet” was used to perform the least absolute shrinkage and selection operator (LASSO) regression analysis on the 9 genes obtained from the TCGA cohort. Finally, four genes were obtained for the model construction. Then, the risk score of each sample was calculated using Equation (1).
riskScore = ∑i = 1nCoef(Xi) × Exp(Xi) (1)
where Coef(Xi) was the corresponding coefficient for each CRG and Exp(Xi) was the expression level of each gene.

The median of risk scores was used as a grouping standard to divide the patients in the TCGA cohort into high-risk (*n* = 180) and low-risk (*n* = 181) groups.

The survival curve was drawn in this study using Kaplan–Meier (K–M) analysis. The area under the receiver operating characteristic (ROC) curve (AUC) was determined to assess the ability of the model to predict the 1-, 2-, and 3-year survival of patients. Based on the expression level of each gene in the model signature, Principal component analysis (PCA) and t-distributed stochastic neighbor embedding (t-SNE) were performed using the R packages “stats” and “Rtsne”, respectively, to observe the distribution of the two groups of people. In addition, univariate and multivariate cox regression analyses were also performed to confirm whether the model could be used as an independent factor for predicting the overall survival (OS) of HCC patients. Moreover, the patients’ risk scores were also calculated based on the RNA-seq data obtained from ICGC and our in-house cohorts to validate the prognostic models. In the ICGC and internal cohorts, the median risk score and 3:1 ratio were selected as cutoff values to classify the patients into low-risk and high-risk groups, respectively.

### 2.4. Functional Enrichment Analysis and Immune Infiltration Score

The DEGs were identified in the high-risk and low-risk groups in the three cohorts using the R package “limma” (fold-change > 1.2 and *p* < 0.05). Then, the common DEGs across the three cohorts were obtained. Gene ontology (GO) analysis of the common DEGs was performed using the R package “clusterprofiler”.

The proportion of 16 immune cells (activated dendritic cell (aDCs), B cells, CD8+ T cells, dendritic cells (DCs), inflammatory dendritic cells (iDCs), macrophages, mast cells, neutrophils, natural killer cells (NK cells), plasmacytoid dendritic cells (pDCs), T-helper cells (Th cells), follicular helper T cell (Tfh), Th1 cells, Th2 cells, tumor infiltrating lymphocytes (TIL), and regulatory cells (Treg)) infiltration and activity levels of 13 typical biological pathways (antigen-presenting cell (APC) co-inhibition, APC co-stimulation, chemotactic cytokines receptor (CCR), check-point, cytolytic activity, human leukocyte antigen (HLA), inflammation-promoting, major histocompatibility complex (MHC) class I, para-inflammation, T-cell co-inhibition, T cell co-stimulation, type I interferon (IFN) response, and type II IFN response) in a single sample were calculated by single-sample gene-set enrichment analysis (ssGSEA) using the R package “gsva” [11].

### 2.5. Single-Cell RNA seq (scRNA-seq) Analysis

The scRNA-seq data of T cells (LIHC_GSE98638 dataset) was downloaded from the Tumor Immune Single-cell Hub (TISCH) database (http://tisch.comp-genomics.org/home/, accessed on 1 June 2022). The scRNA-seq data of tumor tissues, adjacent normal liver tissues, and peripheral blood of six HCC patients were obtained [12]. Based on TISCH, the analysis of DEGs and identification of immune cell types were performed by analyzing scRNA-seq analysis using Seurat [13]. The main steps were as follows. First, an expression matrix of single-cell transcriptome data was acquired, and the cell type information was defined. The single-cell expression matrix was then created as a Seurat object. The FindVariableFeatures function was used to discover hypervariable genes. The data were normalized using the ScaleData function, and the dimensionality reduction clustering was performed using RunPCA and RunUMAP. Finally, the defined cell type information was integrated.

The cuproptosis scores of various T-cell subtypes for prognosis-related CRGs-DEGs were obtained using the ssGSEA package. First, the counts were extracted from the single-cell RNA-seq expression matrix, and the genes with 0 expressions were removed. The GSVA package was then used to calculate the score of the prognosis-related CRGs-DEGs in the extracted counts and added to the Seurat object using the AddMetaData function. Finally, the VlnPlot function was used for visualization.

Similarly, based on the hallmark gene set analysis, the signaling pathway scores of different T cell subtypes were obtained.

### 2.6. Cell Culturing

Seven HCC cell lines (HCCLM3, SNU-449, Huh-7, HCC97-H, HCC97-L, Hep3B) and the human normal liver cell line (MIHA) were used in this study. The Huh-7 and Hep3B were obtained from the Cell Bank of the Chinese Academy of Sciences (Shanghai, China). The MIHA, HCCLM3, SNU-449, HCC97-H, and HCC97-Lwere purchased from Zhong Qiao Xin Zhou Biotechnology (Shanghai, China). All cell lines were cultured in the medium containing 89% DMEM (GIBCO, 11960044), 10% FBS (GBICO, 10099-141), 100U/mL streptomycin, and 100 μ G/mL penicillin (Solarbio, Beijing, China, P1400) in a humidified incubator with 5% CO_2_ at 37 °C.

### 2.7. Multiplex Immunofluorescence Staining

A TSA fluorescence kit was used to process multi-color immunofluorescence staining following the manufacturer’s instructions. First, the primary antibody was incubated with the tissue sections. The following antibodies were used: CD4 (Abcam, Shanghai, China, ab133616; 1:3000), Foxp3 (Servicebio, Wuhan, China, GB112325; 1:1000), CD68 (Servicebio, Wuhan, China, GB113150; 1:3000), and CDKN2A (Servicebio, Wuhan, China, GB111143; 1:100). The tissues were then incubated with the respective polymer horseradish peroxidase (HRP)-E-conjugated secondary antibodies. Then, the tissue sections were heated using a microwave. After all the antigens were labeled, the tissue sections were stained with 4‘-6′-diamino-2-phenylindole (DAPI; Servicebio, G1012). Multi-spectral images were obtained by staining the slides, followed by scanning with a scanner (Pannoramic MIDI: 3Dhistech, Shandong, China). Then, CaseViewer 2.4 software was used to analyze the scanning results and output of multi-color fluorescent images.

### 2.8. Real-Time Polymerase Chain Reaction (PCR)

The total RNAs of hepatocyte and hepatocellular carcinoma cell lines were extracted with TRIzol reagent (Servicebio) following the manufacturer’s instructions. Then, the extracted RNA was reverse-transcribed into cDNA using a PrimeSciptTM RT reagent Kit with gDNA Eraser (Takara, Beijing, China). FastStart Universal SYBR Green Master (Roche, Shanghai, China) was then used for real-time fluorescence quantitative PCR analysis. The primer sequences were as follows: H-CDKN2A-F, 5ʹ-GGAGGCCGATCCAGGTCAT-3ʹ and H-CDKN2A-R, 5ʹ-CACCAGCGTGTCCAGGAAG-3ʹ; H-ACTB-F GTCATTCCAAATATGAGATGCGT, H-ACTB-R GCTATCACCTCCCCTGTGTG. In order to compare the expression levels of different samples, the relative expression level of the gene was calculated using the 2^−ΔΔCt^ method. Each experiment was repeated at least three times independently.

### 2.9. Statistical Analyses

First, the gene expression levels in tumor and non-tumor tissues were compared using Student’s *t*-test. K–M analysis and log-rank test were applied for comparing the OS of patients between the high- and low-risk groups. A Chi-square test was performed in this study to compare the clinical differences between the groups in each cohort. The ssGSEA scores of immune cells and pathways between the two groups were compared using the Mann–Whitney test and the *p*-values were adjusted using the Benjamini–Hochberg (BH) method to confirm the significance of the difference. The R software package (version 4.1.2) was used for the statistical analyses in this study. For all the statistical analyses, a *p*-value of less than 0.05 was considered statistically significant. All the *p*-values were two-tailed.

## 3. Results

### 3.1. Prognostic and Differential Analysis of CRGs in the TCGA Cohort

The results showed that the expression levels of most CRGs (16/19, 84.2%) were different between the tumor and non-tumor groups in the HCC patients.

A total of 9 prognosis-related CRGs were obtained by applying univariate Cox regression analysis to 19 CRGs (Figure 2B). At the same time, survival analysis was conducted on these 9 genes, which suggested that the patients with high expression of these genes except *NF2L2* had lower OS; this verified that 8 of these 9 genes had prognostic potential (Appendix A). The intersection of 16 CRGs-DEGs with 9 prognosis-related CRGs identified 9 prognosis-related genes as CRGs-DEGs. Therefore, 9 CRGs-DEGs related to prognosis were finally obtained and used for subsequent analysis (Figure 2A,C, *p* < 0.05 for all genes). In addition, *LIPT1*, *LIPT2*, *PDHA1*, and *DLAT* had close relations and interactions in the interaction network (Figure 2D). The expression of these genes had a certain correlation (Figure 2E).

In addition, the correlations between clinicopathological features and four genes in the TCGA cohort were verified. Four genes were related to clinical manifestation at different degrees. The expression level of *CDKN2A* was higher in the patients with high grade (“G3&G4”) (*p* < 0.05), and the expression levels of GLS, DLAT, and LIPT1 were higher in the patients with a high stage (“III–IV”) (*p* < 0.05); these results were consistent with clinical experience (Appendix A).

### 3.2. Construction and Exploration of Prognostic Models in the TCGA Cohort

A 4-gene prognostic model was obtained by performing a LASSO-Cox regression analysis on the expression profiles of the screened 9 CRGs. The optimal value of the penalty parameter (λ) was selected to determine the signatures of four genes (Table 2). The risk score was calculated using Equation (2).
Risk score = e^(0.602× CDKN2A expression level + 0.466 × GLS expression level + 0.790 × LIPT1 expression level + 1.842 × DLAT expression level)^
(2)

The median cut-off value of the risk score was selected to divide the patients into low-risk (*n* = 181) and high-risk (*n* = 180) groups (Figure 3A). The results showed that, in the high-risk group of the TCGA cohort, there was advanced tumor node metastasis (TNM) stage and high tumor grade (Table 3, *p* < 0.05). Both the PCA and t-SNE maps showed that the patients in the two risk groups were located in different directions (Figure 3B,C). The patients in the low-risk group were more likely to have longer OS as compared to those in the high-risk group (Figure 3D). As shown in Figure 3E, the K–M curve analysis showed that the OS of patients in the low-risk group was higher as compared to that of high-risk group patients *(p* < 0.001). The AUC values of the HCC patients at 1-, 2-, and 3-year OS were 0.745, 0.649, and 0.650, respectively. This indicated that the OS risk score exhibited a good prediction performance for the prognosis of HCC patients (Figure 3F). Meanwhile, these results were validated using the data obtained from the ICGC cohort and our in-house cohort (Figure 4).

In addition, the prognostic potential of this signature in the recurrent TCGA and in-house cohorts was also studied, and the results showed that the high-risk group had worse recurrence-free survival (Appendix A).

### 3.3. Prognostic Potential of the Constructed Four-Gene Signature

Cox regression analysis was performed using the univariate and multivariate analysis of several clinical traits, which were statistically analyzed. The risk scores in the TCGA cohort were correlated with the OS (Figure 5A); therefore, the risk scores were used as an independent prognostic predictor of OS. At the same time, these results were validated using the data obtained from the ICGC cohort and our in-house cohort (Figure 5A,B).

### 3.4. Functional Enrichment Analysis in Three Cohorts

A total of 268 common DEGs were obtained in the high- and low-risk groups in all three cohorts (Figure 6A, *p* < 0.05, fold-change > 1.2 for all genes). GO enrichment analysis was performed using these DEGs to study their role in biological functions and pathways related to the risk score. Surprisingly, these DEGs were significantly enriched in the biological processes, such as nuclear division and organelle division, which are related to cell division. At the same time, the molecular functions of microtubule binding and tubulin binding were also significantly enriched. Moreover, in terms of cellular localization, the DEGs were mainly concentrated in the chromosomal region and spindle (Figure 6B).

### 3.5. Enrichment Score of Immune Cells Infiltration in Three Cohorts

ssGSEA was used to obtain and compare the risk scores for different immune cells, immune functions, and pathways for the identification of correlations between risk scores and immune status.

The comparison of scores between the two groups showed that the score of DC, macrophages, Treg cells, Th2 cells, MHCclass I molecules, and APC co-stimulation were significantly enriched in the high-risk group, while those of the mast cells, cytolytic activity, and type II IFN response significantly decreased (*p* < 0.05, Figure 7A,D). This enrichment of Tregs and macrophages in the patients in the high-risk group was validated in the data obtained from the ICGC cohort and our in-house cohort (Figure 7B,C,E,F).

### 3.6. Immunosuppressive T cell Related to Cuproptosis Based on scRNA-seq

In order to further explore the correlations between cuproptosis and T cells in HCC, the scRNA-seq data of T cells in HCC were analyzed. In the UMAP, we found a good overlap between the different samples suggesting that there was no significant batch effect for these data (Appendix A). The six subtypes of immune cells, including conventional CD4 T cells (CD4Tconv), CD8 T cells (CD8T), exhausted CD8 T cells (CD8Tex), proliferating T cells (Tprolif), regulatory T cells (Treg), and other cells, were identified and visualized (Figure 8A). Furthermore, the results showed that the E2F targets and MYC targets V1 pathways were significantly activated in the Tprolif cells (Figure 8B,C). Then, the scores of the six T-cell subtypes were identified using the nine prognostic-related CRGs-DEGs, among which, the Tprolif cells scored the highest, followed by Treg (Figure 8D). More interestingly, the Tprolif cells expressed the CD4+ T cell markers (CD4), CD8+ T cell markers (CD8A), and a small amount of Treg cell markers (FOXP3 and LAYN) and significantly expressed the immunosuppression markers (HAVCR2, CTLA4, LAG3, TIGIT, and PDCD1) (Figure 8E). This suggested that the poor prognosis of cuproptosis-related HCC patients might be correlated with the expression of these immunosuppression genes in the enriched Tprolif cells.

### 3.7. Validation of Cuproptosis-Related Genes and Immune Cells

In order to further verify these results, qRT-PCR was performed to compare HCC and normal hepatocyte cell lines. The results indicated that the expression level of CDKN2A was high in Hep3B and HCC97-L cells and low in the HCCLM3 and SNU-449 as compared to the normal hepatocyte MIHA (Figure 9A). At the protein level, it was found that CDKN2A was highly expressed in the HCC tumor tissues as compared to the non-tumor tissues using two-color immunofluorescence (Figure 9B). This result was verified using the Human Protein Atlas (HPA) portal (https://www.proteinatlas.org/, accessed on 10 October 2022) (Figure 9C). In addition, the immune cells of the two groups with high and low CRG scores were also compared (each group, *n* = 3). The results showed that Treg and macrophages were significantly enriched in the high-risk group as compared to the low-risk group. These results suggested that the data analysis and some valuable findings were strongly reliable.

## 4. Discussion

Cuproptosis is a new form of cell death and has potential research potential. Therefore, in this study, the differential expression of CRGs in HCC tumor tissues and non-tumor tissues was identified, and the correlations between these DEGs and prognosis were also obtained. Finally, a prognostic model, integrating the four CRGs, was established. Meanwhile, the patients were divided into high-risk and low-risk groups based on the scores of the model. The results showed that the Tregs and macrophages were significantly enriched in the high-risk group. These results were validated using the data obtained from the ICGC cohort and our in-house cohort. Further analysis of the scRNA-seq data revealed that cuproptosis might be correlated with the Tprolif cells, having high expression of immunosuppression markers genes. These conclusions were verified using qRT-PC, multiplex immunofluorescence staining, and IHC.

Currently, different prognosis models of HCC based on a variety of cell death mechanisms, such as ferroptosis or pyroptosis, have been developed [14,15]. Similarly, studies have also demonstrated the potential of CRGs in constructing prognostic models in a variety of tumors, such as clear cell renal cell carcinoma [16]. Therefore, the predictive potential of CRGs in HCC was studied in this study. Surprisingly, 84.2% of the CRGs were DEGs between the HCC tumor and non-tumor tissues. Moreover, based on the phenomenon that half of the CRGs were related to OS, this study suggested the possible impact of cuproptosis on HCC and the possibility of using these CRGs to establish a prognosis model. In particular, the difference analysis showed that 8 of the 9 genes were upregulated in tumor tissues, while *FNE2L2* was downregulated in tumor tissues. *NFE2L2* is considered a tumor suppressor gene [17]. It can inhibit lipogenesis, support β-oxidation of fatty acids, facilitate flux through the pentose phosphate pathway, and increase NADPH regeneration and purine biosynthesis [18]. This might be the reason for the differential expression of *NFE2L2* in the heatmap. This study constructed an HCC prognosis model based on four CRGs, including CDKN2A, GLS, lipt1, and DLAT. CDKN2A is mainly related to the cell cycle and is associated with poor prognosis in multiple cancers [19]. The homozygous deletion of CDKN2A on chromosome 9 was confirmed to be associated with HCC [20]. Glutamase is a key enzyme in glutamine metabolism, converts glutamine into glutamate, and breaks down and produces alpha-ketoglutaric acid, which is then metabolized into the TCA cycle to provide energy [21]. At the same time, glutamine is also involved in the synthesis of glutathione, a natural copper partner in cells; its deletion can cause cuproptosis [4]. Numerous studies showed that GLS was a key gene for the treatment of HCC [22,23]. Lipt1 is a gene involved in the lipoic acid pathway that combines fatty acylation and copper components in the TCA cycle in mitochondria. It is a key cell-death medium that is induced by copper ionophores [4]. DLAT is an enzyme component of the pyruvate dehydrogenase (PDH) complex, which is a target of acylated proteins [4]. PDH can inhibit the TCA cycle [4]. At the same time, the combination of PDH and copper in the acylated TCA cycle might cause DLAT acylation-dependent oligomerization, resulting in cuproptosis [4]. Collectively, these findings suggested that CRGs might affect the development and prognosis of HCC, thereby demonstrating the predictive potential of the four-gene signature model.

In addition, the functional enrichment analysis showed that the CRGs were enriched in pathways that were related to cell division. A previous study reported that copper could participate in regulating cell growth and proliferation as a dynamic signal metal and metal flow regulator, such as mitogen-activated protein kinase kinase 1 (MEK1) and MEK2 [24]. This study suggested that the effects of cellular copper concentration on cell division and proliferation might be a regulating mechanism of cuproptosis in HCC.

Currently, the combination of an immune checkpoint inhibitor (ICI) and vascular endothelial growth factor (VEGF) inhibitor is used as a first-line treatment for advanced liver cancer. The efficacy of ICI treatment depends on TIME; therefore, the TIME of HCC is very important for its advanced treatment [25]. A previous study reported that ferroptosis was associated with the TIME of HCC [14]. The patients with high scores in the ferroptosis model showed poor anti-tumor immune function and a worse prognosis. The low-risk group had a better prognosis to predict the OS compared to that of the high-risk group. In all three cohorts, the proportion of macrophages and Tregs in the high-risk group was higher as compared to that in the low-risk group. The poor prognosis of HCC patients was correlated with the increase in the proportion of Tregs or tumor-associated macrophages [26,27,28], which further verified the predictive potential of this model. Moreover, the cytolytic activity score of the low-risk group was high, and the HCCs with high cytolytic activity were associated with enhanced immunity and better survival [29], which also verified the accuracy of the model prediction results. It was noteworthy that the APC co-stimulatory score and MHC class I score in the low-risk group were lower compared to those in the high-risk group. This indicated that there were differences in the antigen presentation and T cell activation processes between the high- and low-risk groups. This might be one of the specific mechanisms of cuproptosis in HCC. Additionally, the scRNA-seq analysis of T cells obtained from HCC patients in this study was also performed. The results showed that the Tprolif and Treg cell subsets were the most related cells to cuproptosis. It was worth noting that the Tprolif cells could highly express HAVCR2, CTLA4, LAG3, TIGIT, and PDCD1, which are the markers related to immune exhaustion. Zhang et al. [26] also showed that the same group of MKI67+ CD8+ T cells could transform the developmental direction of depleted CD8+ T cells. In melanoma, a study [30] showed that this group of cells was the most profound in the earlier stages of the dysfunctional program in the CD8+ T cells. Moreover, this group of cells could also cause the dysfunction of CD8+ T cells, including the depletion of T cells [30]. The lymphoid cells in ascites also contain these proliferative T cells, which might be due to the migration of tumor tissue to the abdominal cavity, resulting in potential immunosuppression of the tumor [26]. These studies showed that the HCC patients with cuproptosis had a poor prognosis, which might be related to the expression of these immunosuppressive genes in their enriched Tprolif cells. These findings provided new insights into the effective combination of immunotherapy for HCC patients.

In the latest study [31], it was found that the expression levels of CRGs in tumor and normal liver tissues were significantly different and were significantly correlated with the poor survival rate of HCC. This indicated that CRGs might become an important marker and new target for the early diagnosis, precise treatment, and prognostic evaluation of HCC. At present, studies on cuproptosis in HCC have been carried out in succession. Irrespective of the scoring model based on CRGs, cuproptosis-related lncRNAs, or cuproptosis-related miRNAs in previous studies, they have obtained similar results as were obtained in the current study: the patients with higher cuproptosis-related scores had lower OS [32,33,34,35,36]. In addition, the drug sensitivity analyses conducted by Lei Ding et al. [32] and Qiongyue Zhang et al. [35] suggested that the sample with high cuproptosis-related scores had lower IC50. This indicated that the sample with a high cuproptosis-related score might benefit more from most types of chemotherapies. The advantage of this study was establishing a four-genes-based model using an algorithm, which was more conducive to the clinical study. In addition, this study not only used the public datasets to establish the model and verify the model but also used the public datasets and our in-house cohort to verify the results, which greatly improved the reliability of the data. At the same time, qRT-PCR and multiple immunofluorescence experiments were performed to verify the results at the mRNA and protein levels, which further enhanced the reliability of the study. Furthermore, this study analyzed the immune infiltration related to cuproptosis at a single-cell level, which was helpful to explain the mechanism of cuproptosis in HCC. Nevertheless, there were certain limitations to this study as well. First, CRGs are continuously being explored; therefore, only 19 genes were included, which might not be comprehensive enough. Second, only one marker was considered for the establishment of the prognostic model; therefore, many important prognostic genes of HCC might not be included. Finally, the correlation between risk score and immune activity was not experimentally addressed before and requires further studies.

## 5. Conclusions

In conclusion, a new HCC prediction model, containing four CRGs, was constructed based on cuproptosis. The model showed an independent association with OS in all three cohorts, thereby showing the potential to predict the prognosis of HCC. Moreover, this study also revealed that the cuproptosis-related HCC was significantly enriched in the Treg cells and macrophages and was closely related to the depletion of proliferative T cells, thereby providing a basis for the combined immunotherapy of HCC.

## Figures and Tables

**Figure 1 cancers-14-05713-f001:**
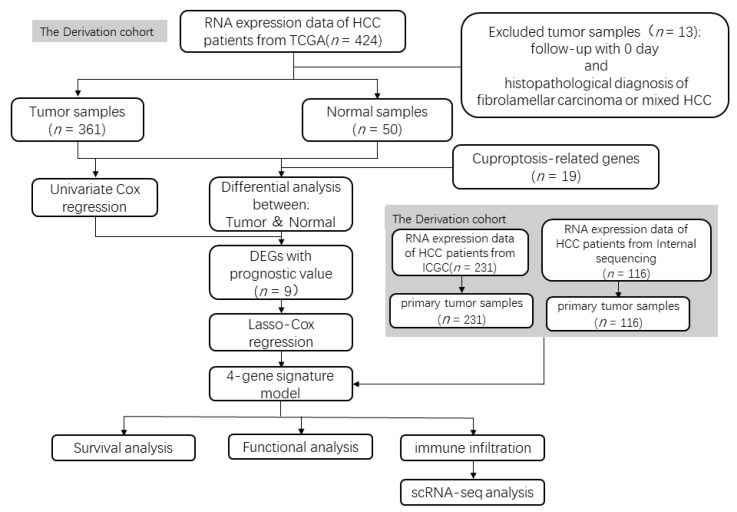
Flow chart of patients’ enrollment and data processing. ‘*n*’ means the number of samples or genes.

**Figure 2 cancers-14-05713-f002:**
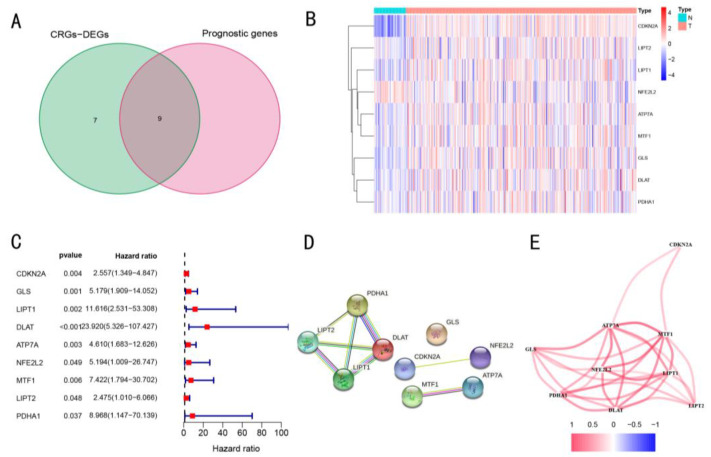
Identification of CRGs for follow-up studies. (**A**) Venn diagram of the intersection of CRGs-DEGs and cuproptosis-related prognostic genes in tumor and non-tumor tissues. (**B**) The heatmap shows the comparison of the expression amount of intersection genes in tumor (T) and non-tumor (N) tissues. (**C**) Forest plot shows the risk degree of each intersecting gene using the univariate cox regression analysis. (**D**) Interactions between the nine genes were presented using the PPI network for further study. (**E**) Correlation network shows the correlations between the expression levels of nine genes with positive correlation shown in warm colors.

**Figure 3 cancers-14-05713-f003:**
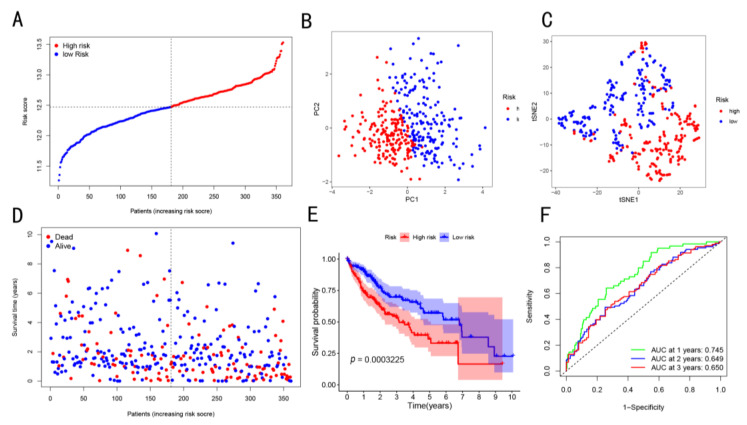
Four-gene signature model was constructed using the TCGA cohort data. (**A**) Risk curves show the distribution of sample risk values. (**B**) PCA analysis of high- and low-risk groups. (**C**) t-SNE analysis of two groups was performed. (**D**) Survival status vs. time. The graph shows that with the increase in risk, the number of deaths increased. (**E**) K–M curve for the two groups. (**F**) Time-dependent ROC curve of risk score prognostic performance.

**Figure 4 cancers-14-05713-f004:**
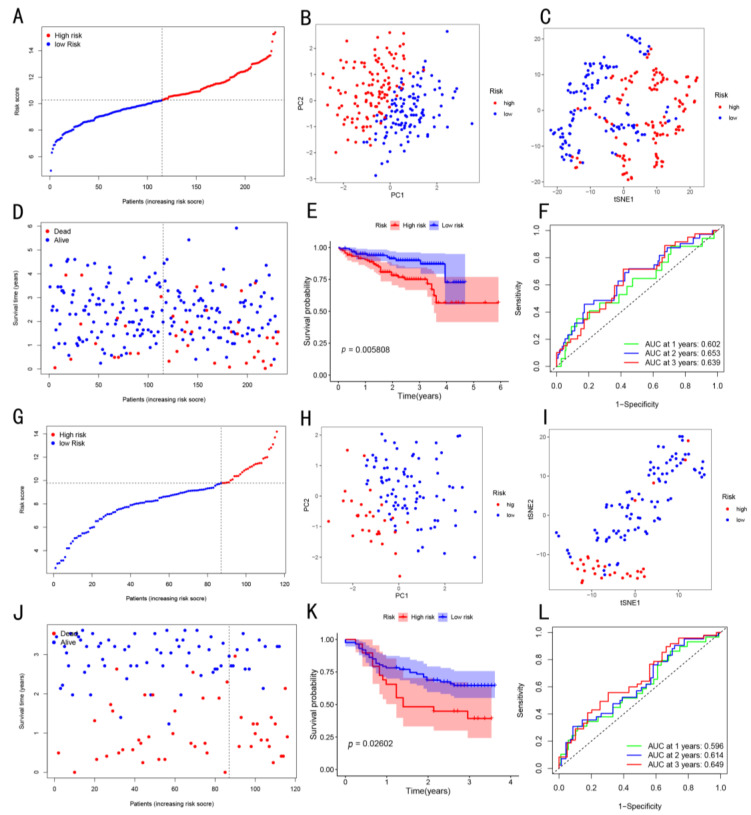
Validation of the four-gene signature using the ICGC cohort data and our in-house cohort data (named as an in-house cohort). (**A**–**F**) and (**G**–**L**) are the analyses of the ICGC cohort and our in-house cohort data, respectively. (**A**) Risk curves, showing the distribution of sample risk values in the ICGC cohort. (**B**) PCA analysis of high- and low-risk groups in the ICGC cohort. (**C**) t-SNE analysis of two groups performed in the ICGC cohort. (**D**) Survival status vs. time in the ICGC cohort. (**E**) K–M curve for the two groups of patients in the ICGC cohort. (**F**) Time-dependent ROC curve of risk score prognostic performance in the ICGC cohort. (**G**) Risk curves, showing the distribution of sample risk values in the in-house cohort. (**H**) PCA analysis of the high- and low-risk groups in the in-house cohort. (**I**) t-SNE analysis of two groups performed in the in-house cohort. (**J**) Survival status vs. time in the in-house cohort. (**K**) K–M curve for two groups of patients in the in-house cohort. (**L**) Time-dependent ROC curve of risk score prognostic performance in the in-house cohort.

**Figure 5 cancers-14-05713-f005:**
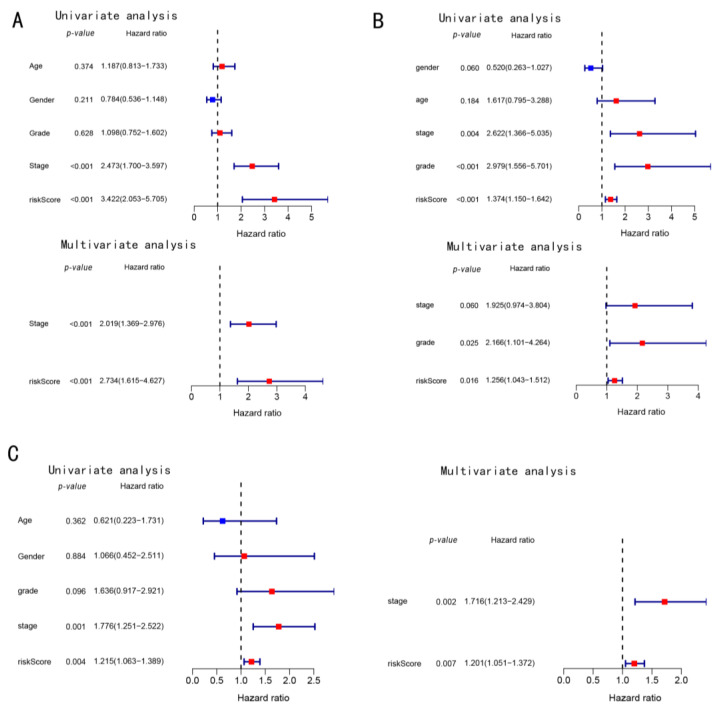
Univariate and multivariate Cox regression analysis of the (**A**) TCGA cohort data, (**B**) ICGC cohort, and (**C**) in-house cohort data.

**Figure 6 cancers-14-05713-f006:**
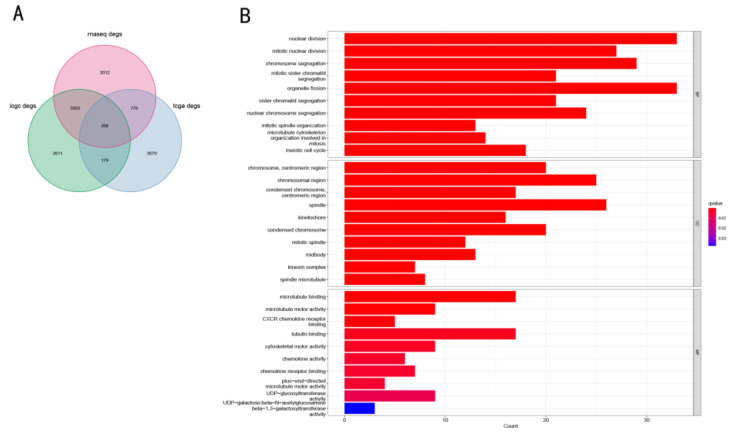
GO analysis results of common DEGs in the three cohorts. (**A**) Venn diagram of the number of genes included in the GO analysis. *p* < 0.05 and fold-change > 1.2. (**B**) GO analysis of the common DEGs.

**Figure 7 cancers-14-05713-f007:**
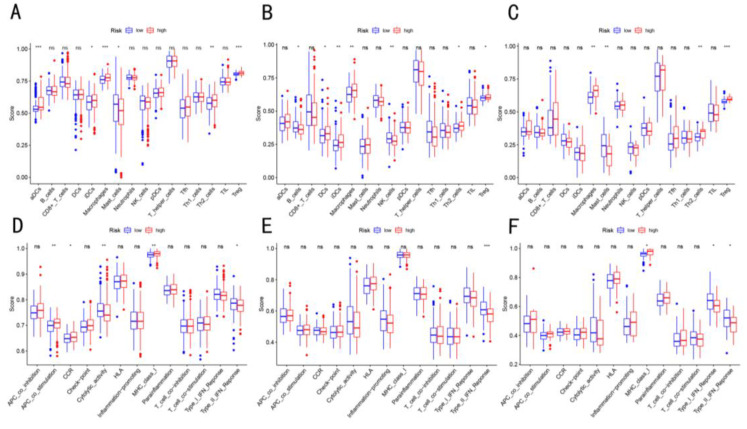
Enrichment scores of immune cell infiltration in the three cohorts. The comparison of the enrichment scores of immune cell infiltration between two risk groups in (**A**,**D**) TCGA cohort, (**B**,**E**) ICGC cohort, and (**C**,**F**) in-house cohort. (**A**–**C**) Boxplots show scores for multiple immune cells. (**D**–**F**) Boxplots show scores for multiple immune-related functions. ns, not significant; * *p* < 0.05; ** *p* < 0.01; *** *p* < 0.001.

**Figure 8 cancers-14-05713-f008:**
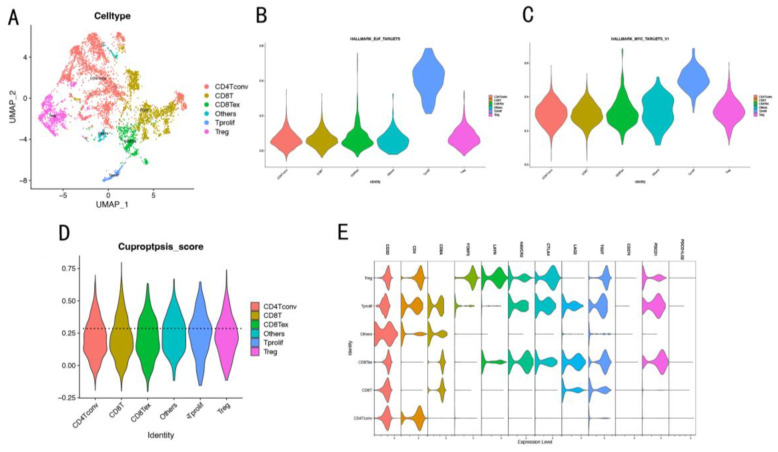
Tprolif cells associated with cuproptosis in the HCC were obtained based on scRNA-seq. (**A**) UMAP diagram shows the T cell subtypes of the patients. Each cluster is color-coded according to the cell type, and the cluster annotations are shown in the figure. (**B**,**C**) E2F and MYC target V1 pathways are significantly enriched in the Tprolif subgroup based on the hallmark gene set using the GSVA.(**D**) Scores of CRGs in T cell subsets. (**E**) Violin diagram shows the expression characteristics of 12 marker genes in T cell subtypes.

**Figure 9 cancers-14-05713-f009:**
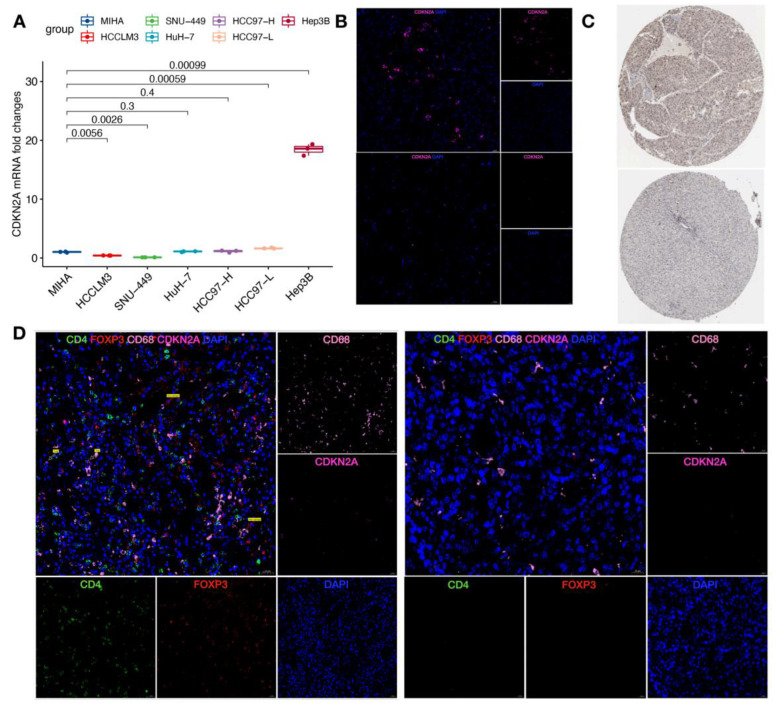
Validation of the CRGs and immune cells. (**A**) The expression of CDKN2A between MIHA as a normal hepatocyte line and 6 HCC cell lines was analyzed by qRT-PCR. (**B**) Two-color immunofluorescence results showed that CDKN2A was highly expressed in tumor tissues (upper panel) compared with non-tumor tissues (lower panel). (**C**) immunohistochemistry in the HPA showed a difference in CDKN2A expression in HCC (upper panel) and non-cancer issues (lower panel). (**D**) Multicolor immunofluorescence confirmed that the concentration of Treg and macrophages with a high-risk group (left panel) was higher than that in the low-risk group (right panel).

**Table 1 cancers-14-05713-t001:** Clinical information of the HCC patients included in this study.

	TCGA	ICGC (LIRI-JP)	In-House RNA-seq
Number of patients	361	231	116
Age (median, range)	61 (16–90)	69 (31–89)	49.5 (36–76)
Gender			
Female	117	61	15
Male	244	170	101
Grade			
Grade 1–2	224	53	15
Grade 3–4	132	159	101
unknown	5	19	0
AFP			
≤200	196	NA	63
>200	75	NA	53
unknown	90	NA	0
Stage			
Stage I	167	36	58
Stage II	82	105	32
Stage III	84	71	26
Stage IV	4	19	0
unknown	24	0	0
vascular_tumor			
Macro	16	NA	23
Micro	89	NA	49
None	200	NA	44
unknown	56	NA	NA
Survival status			
OS days	588 (0–3675)	780 (10–2160)	945 (0–1320)

**Table 2 cancers-14-05713-t002:** Screened genes.

Cuproptosis-Related Genes	CRGs-DEGs	Prognosis Value	Intersect Genes	Signature Model
NFE2L2	√	√	√	
NLRP3	√			
ATP7B				
ATP7A	√	√	√	
SLC31A1	√			
FDX1	√			
LIAS	√			
LIPT1	√	√	√	√
LIPT2	√	√	√	
DLD				
DLAT	√	√	√	√
PDHA1	√	√	√	
PDHB	√			
MTF1	√	√	√	
GLS	√	√	√	√
CDKN2A	√	√	√	√
DBT	√			
GCSH	√			
DLST				

**Table 3 cancers-14-05713-t003:** Clinical information of the patients in high- and low-risk groups in the three cohorts.

Characteristics	TCGA-LIHC Cohort			ICGC-LIRP-JP Cohort			Internal RNA-seq Cohort		
	High Risk	Low Risk	*p*-Value	High Risk	Low Risk	*p*-Value	High Risk	Low Risk	*p*-Value
Gender			0.76348			0.43206			0.63174
Female	57	60		28	33		5	10	
Male	123	121		88	82		24	77	
Age			0.72431			0.46656			0.20261
≤65 year	113	111		42	47		28	70	
>65 year	66	70		74	68		1	12	
unknown	1	0		0	0		0	0	
Grade			0.00845			0.01173			-
G1 and G2	100	124		69	84		29	86	
G3 and G4	78	54		38	21		0	0	
unknown	2	3		9	10		0	1	
AFP			0.97195			-			0.45128
≤200	91	105		-	-		14	49	
>200	35	40		-	-		15	38	
unknown	54	36		-	-		0	0	
vascular_tumor			0.73062			-			0.18493
Yes	51	54		-	-		21	51	
no	93	107		-	-		8	36	
unknown	36	20		-	-		-	-	
Stage			0.00136			0.11724			0.44054
I and II	109	140		65	76		24	66	
III and IV	56	32		51	39		5	21	
unknown	15	9		0	0		0	0	

## Data Availability

All datasets analyzed during the current study are available from the corresponding author upon reasonable request.

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
