# Peer review of "Comprehensive Analysis of Cuproptosis-Related Genes in Prognosis and Immune Infiltration of Hepatocellular Carcinoma Based on Bulk and Single-Cell RNA Sequencing Data"

_cancers, 2022, doi:10.3390/cancers14225713_

Round 1

Reviewer 1 Report (Previous Reviewer 3)

The authors have revised their manuscript based on my previous critiques. The newly added data along with significant text modifications have improved this version. Hence, the manuscript can be judged for publication at this stage.

Reviewer 2 Report (Previous Reviewer 2)

After revision, manuscript looks in shape of publication. Accept manuscript after minor changes in the text and formatting..

This manuscript is a resubmission of an earlier submission. The following is a list of the peer review reports and author responses from that submission.

Round 1

Reviewer 1 Report

Cuproptosis is a novel pathway in the last 2 years, with emerging papers in September 2022 on HCC, CRC, melanoma.

PMID: 36177029, 36159561, 36153416, 36147869, 36147507, ...etc on HCC could be screened and compared with the current analysis or complement if possible in case the main focus was lncRNAs or miRNAs.

Section 2.3 needs expansion and more details, e. which 3 cohorts, which immune cells? the cutoff value of risk score for high/low groups?

Similarly, section 2.5 is unclear and needs details.

The formula Line 182, needs to be revised and completed.

Did authors made any validation for the deregulated gene based on sex, race, age, ...etc?

Reviewer 2 Report

In this manuscript Yang et al., authors reported unique relationship of cuproptosis-related genes in hepatocarcinoma. It has been previously reported about unique cellular death phenomenon “cuproptosis”. Using bio-informatics approach, authors suggested its unique role in tumor prognosis. In this study, author uses mainly bio-informatics approach i.e., bulk and single cell RNA sequencing to answer their research goal. Datasets were obtained from different platform, TCGA, ICGC and in-house cohorts.

The manuscript proposes a unique finding but need minor revision and explanation prior to publication.

Comments

Minor Comments:

1.     Figure 2A, author has shown the venn-diagram and compared DEGs (between tumor vs non-tumor) with prognostic genes. It would be appreciated to explain the source of these prognostic genes. How many genes were considered as a prognostic marker?

2.     Once, datasets were compared between tumor vs non-tumor. It seems that author found only 16 gene in DEGs. Is that true? Or it has been filtered out based on some internal criteria (P-value and logFC).

3.     Authors mentioned risk score across the samples using 4 selective genes. Would it be enough to evaluate their findings or needs to be added more genes i.e., either from network analysis or increasing broader criteria to add more valuable samples? If not, I would suggest showing findings where others have used similar criteria support their research goal.

4.     In the risk score calculation, how author calculated a gene coefficient? I assume that risk scores were summed for all 6 cuproptosis related genes to calculate final score/ sample.

5.     In their study, larger differences in the sample’s numbers were found between tumor (361) vs non-tumor (50). Have author used downsample approach to equalize the sample number on random basis and perform similar analysis?

6.     There is no explanation of figure 2B heatmap. What is obvious that authors want to show the relationship between the genes across two groups. Author did not comment on the high expression of NFE2L2 genes in the normal samples while other markers make sense.

7.     LASSO regression analysis was performed to avoid overfitting of 9 genes across samples and further analysis leads to 4 selected genes for and generated “Prognostic model”. Here, author mentioned “λ” without defining what this stands for.

8.     How were risk scores calculated in figure 5 and why their calculations were different from previous figures?

9.     Have author kept the DEGs filtering criteria same for all of their samples before looking for shared gene population between three datasets (TCGA, ICGC and In-house).

10.   In the scRNAseq analysis, authors have used around 5-6 patients to look for immune status using 9 cuproptosis related genes. But author did not mention type of integration and U-MAP to show good sample overlapping or no batch effect. This figure may go to supplementary but would be valuable for readers.

11.  Manuscript needs more background explanation in material and method section, but I liked the content of research.

12.  Author should provide code for prognostic model evaluation from their three cohorts and other R code used in the article, to review and replicate at our end. Sometimes, methods are not very descriptive to understand.

13.  I am assuming that author would also submit their data to Github repository. 

Reviewer 3 Report

In this manuscript, Yang and colleagues set out to identify the prognostic value of 9 genes associated with cuproptosis, a newly identified form of copper dependent programmed cell death. Using cox analysis and other statistical parameters, they reveal that this cuproptosis related signature could offer potential in determining survival kinetics amongst liver cancer patients. Further to this, they also show that regulatory T cells and macrophages could be enriched alongwith cuproptosis-related signatures via their analysis of single-cell sequencing of publicly available liver cancer datasets. Overall, their statistical and bioinformatic analysis appear sound. The paper provides bioinformatic analysis justification for the prognostic value of cuproptosis related genes, however, the biological insights of these gene-signature in the HCC TME is not validated.

Major comments:

1) My concern with this paper remains that these genes are not biologically validated in HCC tumor specimens or in cell-lines. The fact that some of these genes have a role in Cu-dependent cell death is also not shown.

2) Fig 3: in addition to overall survival, have the authors looked into the prognostic value of this signature in recurrent HCC datasets?

3) A clinical confirmation of some of the cu-related genes with TRegs/macrophages within the HCC tumors will be required to fully support their dry-lab analysis.

4) Minor: there are many typos in this version. Please exercise more care in drafting the manuscript.
